# Feasibility of Endotracheal Extubation Evaluation Form in Predicting Successful Extubation in Neonatal Intensive Care Units: A Retrospective Study

**DOI:** 10.3390/children10061053

**Published:** 2023-06-13

**Authors:** Yung-Cheng Liu, Ching-Yi Yeh, Shu-Ting Yang, Wei-Chan Chung, Tuan-Jung Hsu, Chau-Chyun Sheu, Hsiu-Lin Chen

**Affiliations:** 1Division of Respiratory Therapy, Kaohsiung Medical University Hospital, Kaohsiung 807, Taiwan; willian7910@gmail.com (Y.-C.L.); beer940829@yahoo.com.tw (C.-Y.Y.); u97009045@gmail.com (W.-C.C.); hsutuanjung@gmail.com (T.-J.H.); 2Department of Pediatrics, Kaohsiung Medical University Hospital, Kaohsiung 807, Taiwan; 1000410@kmuh.org.tw; 3Division of Pulmonary and Critical Care Medicine, Department of Internal Medicine, Kaohsiung Medical University Hospital, Kaohsiung 807, Taiwan; sheucc@gmail.com; 4Department of Internal Medicine, College of Medicine, Kaohsiung Medical University, Kaohsiung 807, Taiwan; 5Department of Respiratory Therapy, College of Medicine, Kaohsiung Medical University, Kaohsiung 807, Taiwan

**Keywords:** airway extubation, mechanical ventilation, preterm infants, form, intensive care units, ventilator weaning

## Abstract

Given the limited availability of evidence-based methods for assessing the timing of extubation in intubated preterm infants, we aimed to standardize the extubation protocol in this single-center, retrospective study. To accomplish this, we established an extubation evaluation form to assess the suitability of extubation in preterm infants. The form comprises six indicators: improved clinical condition, spontaneous breath rate ≥ 30 breaths per minute, peak inspiratory pressure (PIP) ≤ 15 cmH_2_O, fraction of inspired oxygen (FiO_2_) ≤ 30%, blood pH ≥ 7.2, and mixed venous carbon dioxide tension (PvCO_2_) < 70 mmHg. Each positive answer is given one point, indicating a maximum of six points. We enrolled 41 intubated preterm infants (gestational age < 32 weeks, birth weight < 1500 g) who were receiving mechanical ventilation support for over 24 h. Among them, 35 were successfully extubated, and 6 were not. After completing the extubation evaluation form and adjusting for birth weight and postextubation device, we observed that the total score of the form was significantly associated with successful extubation; the higher the score, the greater the chance of successful extubation. Thus, we infer that the extubation evaluation form may provide a more objective standard for extubation assessment in preterm infants.

## 1. Introduction

Preterm infants often require intubation with mechanical ventilation support during respiratory failure treatment [1]. However, for intubated preterm infants, no consensus has been reached as to the most suitable methods for weaning mechanical ventilation. The methods for assessing suitability of extubation in preterm infants remain primarily subjective and depend on the clinical procedures provided by the institution or the personal training and experience of the attending physicians. In addition, clinicians usually consider biochemical data, clinical condition, and the setting of mechanical ventilation when making clinical decisions regarding extubation [2]. These factors may result in critical differences in clinical assessments of patients.

A study of preterm infants with extremely low birth weights revealed a successful extubation rate of merely 60% [3]. Furthermore, a systematic review conducted in 2014 on preterm infants with gestational age < 32 weeks or birth weight < 1500 g stated that the rate of these infants being reintubated was 25% ± 9% (mean ± standard deviation) [4].

Spontaneous breathing trials, which have been regularly applied in assessing the extubation readiness of adults, lack the evidence necessary to be performed in preterm infants [5]. Hermeto et al. discovered that the application of a weaning protocol can reduce the time required for the use and weaning of mechanical ventilation; however, the study presented limited evidence [3]. A literature review conducted in 2016 failed to find evidence of a difference between protocolized and nonprotocolized weaning methods in reducing the duration of mechanical ventilation in preterm infants [6]. This lack of relevant data indicates that studies on this type of patients are few, and additional studies on extubation in preterm infants are required if a protocol is to be established for the successful weaning off of mechanical ventilation in these patients.

In preterm infants, delayed extubation may result in a longer duration of mechanical ventilation and increase the risks of severe bronchopulmonary dysplasia (BPD) and neurodevelopmental disability [7,8]. Furthermore, compared with preterm infants with successful extubation, those with failed extubation have higher risks of BPD and mortality; in addition, they require mechanical ventilation and hospitalization for longer periods [9,10]. Because the timing for performing extubation in preterm infants is delicate, correctly predicting successful extubation in this population has been a topic of major concern for neonatology teams.

The purpose of this study is to standardize the protocol for extubation in preterm infants and reduce the amount of subjective influence in assessments of the conditions for extubation. We aim to objectively assess and predict the most suitable time for endotracheal extubation in preterm infants to reduce mortality and incidence of complications and, thereby, improve their subsequent development and quality of life.

In this retrospective study, we analyzed the relevant factors in successful extubation in preterm infants and determined whether the extubation evaluation form developed by the neonatal intensive care unit of Kaohsiung Medical University Chung-Ho Memorial Hospital can effectively predict successful extubation.

## 2. Materials and Methods

This retrospective study was reviewed and approved by the Institutional Review Board of Kaohsiung Medical University Chung-Ho Memorial Hospital (approval number: KMUHIRB-SV(I)-20200029, date: 30 June 2020). This project was drafted in accordance with the Human Subjects Research Act of Taiwan.

This study adopted a retrospective research design and was performed at Kaohsiung Medical University Chung-Ho Memorial Hospital from 1 January 2014 to 24 April 2019. Preterm infants were eligible for the study if they had (1) gestational age less than 32 weeks and (2) birth weight less than 1500 g. PSIMV + PS mode was used before extubation [11]. Those who had accidental extubation, extubation from high-frequency oscillatory ventilation, death before extubation, intubation due to surgery, mechanical ventilation via an endotracheal tube for less than 24 h, or missing data were excluded from the analysis.

Based on the results of their extubation, the subjects were divided into two groups: specifically extubation success (ES) and extubation failure (EF) groups. ES was defined as reintubation not being required within 72 h of the first extubation attempt, whereas EF was defined as endotracheal reintubation being required within 72 h of the first extubation attempt [4]. BPD was classified according to the 2001 NICHD consensus workshop [12]. Retinopathy of prematurity (ROP) was classified according to The International Classification of Retinopathy of Prematurity [13].

Reintubation was considered necessary if the patients experienced any of the following: (1) PvCO_2_ greater than 65 mmHg with blood pH lower than 7.2; (2) continuous use of FiO_2_ at a concentration greater than 60% to sustain SpO_2_ at 88–92%; (3) frequent (more than 3–8 times per hour) apnea or severe apnea requiring positive pressure ventilation.

Because the decision to reintubate is made by the responsible physician and the conditions of preterm infants change rapidly and diversely, situations requiring reintubation may differ from the aforementioned standard.

We assessed the effectiveness of the form in predicting successful extubation by using statistical analysis of the extubation results and the scores obtained from the extubation evaluation form. The extubation evaluation form is shown in Figure 1.

The extubation indicators included in the extubation evaluation form were the following: (1) improved clinical condition: physicians and respiratory therapists would conduct daily integrated assessment of whether the principal diagnosis, cause of mechanical ventilation requirement, and chest X-ray had improved (e.g., whether the stage of respiratory distress syndrome improved) as well as whether the C-reactive protein level had decreased; (2) spontaneous breath rate ≥ 30 breaths per minute (bpm); (3) PIP ≤ 15 cmH_2_O; (4) FiO_2_ ≤ 30%; (5) blood pH ≥ 7.2: the time of reference is 48 h before extubation to the moment of extubation; (6) PvCO_2_ < 70 mmHg: the time of reference is 48 h before extubation to the moment of extubation [14].

The extubation evaluation form consisted of the six presented indicators. One point was assigned for each indicator a patient met, with a total of six points possible. According to its design, when a patients’ total score is ≥4, extubation is recommended. However, in this study, attending physicians were not required to follow this evaluation form; they were given the option to use their own judgement to decide when to extubate in their patients.

Statistical analyses were performed using SPSS Statistics version 20 (IBM, Armonk, NY, USA) and JMP version 10 (SAS Institute, Cary, NC, USA). All tests were two-tailed, with significance set at *p* < 0.05.

Fisher’s exact test and Wilcoxon’s rank-sum test were used to compare the characteristics of the ES and EF groups. The results are presented as frequency, median, percentage, and interquartile range. Logistic regression was used to analyze the factors associated with successful extubation. Receiver operating characteristic (ROC) curve analysis was used to analyze the cut-off point, sensitivity, and specificity and to examine the prediction effectiveness of the extubation evaluation form.

## 3. Results

A total of 41 subjects admitted from 1 January 2014 to 24 April 2019 met the inclusion criteria. Among them, 35 were successfully extubated and 6 had failed extubation, indicating a failure rate of 14.6%. The inclusion process is presented in Figure 2.

During the perinatal period, several variables did not significantly differ between the ES and EF groups, such as gestational age, gender, being born extremely preterm, being born with an extremely low birth weight, antenatal steroid administration, vaginal delivery, Apgar score at 5 min, Neonatal Therapeutic Intervention Scoring System score, surfactant use, aminophylline or caffeine use, patent ductus arteriosus, late-onset sepsis, and the administration of postextubation nasal intermittent positive pressure ventilation. The only exception was birth weight, with the ES group being significantly heavier than the EF group (971 g vs. 750 g, *p* = 0.015).

During the first extubation attempt, preterm infants in the ES group had significantly low peak inspiratory pressure (13 cmH_2_O vs. 16 cmH_2_O, *p* = 0.006), FiO_2_ (25% vs. 40%, *p* = 0.001), and set respiratory rate (25 bpm vs. 40 bpm, *p* = 0.042). However, other variables, including days of life, postmenstrual age, body weight, and blood pH and PvCO_2_, did not significantly differ between the ES and EF groups. The results are organized in Table 1.

The results obtained indicated that, compared with the ES group, the EF group had significantly longer cumulative mechanical ventilation (5 days vs. 51 days, *p* = 0.002), respiratory support device use (including the use of invasive and noninvasive positive pressure ventilation, the administration of nasal continuous positive airway pressure, and the use of high-flow nasal cannula) (72 days vs. 110 days, *p* = 0.009), and ICU (73 days vs. 112 days, *p* = 0.011) and hospital stays (93 days vs. 133 days, *p* = 0.009).

With regard to complications, compared with the ES group, the EF group had significantly higher rates of developing ROP grade 3 or higher (8.6% vs. 83.3%, *p* < 0.001) and moderate or severe BPD (40% vs. 100%, *p* = 0.009). The results are presented in Table 2.

Preterm infants may need to be reintubated for multiple reasons. In the EF group, reintubation was necessary due to apnea (17%), bradycardia (33%), desaturation (83%), respiratory acidosis (50%), and increased work of breathing (50%).

Three patients in the EF group experienced problems with their endotracheal tubes, such as leakage or difficulty maintaining proper position. As a result, extubation and noninvasive positive pressure ventilation were attempted for these infants, leading to a higher setting of mechanical ventilation before extubation in the EF group.

We performed a Wilcoxon rank-sum test and a Fisher exact test to compare the total scores (median and interquartile range) of the two groups and the percentage of participants scoring positively on each indicator. The results are displayed in Table 3. Significantly, more participants in the ES group scored positively for the FiO_2_ ≤ 30% (85.7% vs. 16.7%, *p* = 0.002) and PIP ≤ 15 cmH_2_O (80% vs. 33.3%, *p* = 0.035) indicators than those of the EF group did. For improved clinical conditions, blood pH ≥ 7.2, spontaneous breath rate ≥ 30 bpm, and PvCO_2_ < 70 mmHg, no significant differences were observed between the two groups. The ES group had a significantly higher final total score than the EF group did (median 6 vs. 4, *p* = 0.001)

A univariate logistic regression revealed that the parameters associated with successful extubation were birth weight (odds ratio (OR): 1.007, 95% CI: 1.001–1.014), FiO_2_ before extubation (OR: 0.707, 95% CI: 0.548–0.912), PIP before extubation (OR: 0.496, 95% CI: 0.27–0.911), set respiratory rate before extubation (OR: 0.92, 95% CI: 0.852–0.993), and total score of the extubation evaluation form (OR: 5.187, 95% CI: 1.589–16.934). Other parameters, such as gestational age, postextubation device, pre-extubation days of life, pre-extubation postmenstrual age, and pre-extubation body weight did not show significance in the univariate logistic regression analysis of the present study. Nonetheless, they were included in the multivariate logistic regression analysis because they were considered critical factors in previous studies [15,16].

The aforementioned parameters were input into the multivariate logistic regression. After colinear combinations were removed and birth weight and postextubation device were adjusted for, the total score of the extubation evaluation form still exhibited a significant association with successful extubation (OR: 6.649, 95% CI: 1.297–34.075). This indicates that extubation evaluation form scores can predict whether extubation will be successful, with a high score denoting a higher chance of success. The results are presented in Table 4.

Subsequently, ROC analysis was performed to obtain a prediction model for the total score of the extubation evaluation form after the birth weight and postextubation device were adjusted for. The area under curve (AUC) was 0.938 (Figure 3).

The extubation evaluation form was then applied to the study group. The total score of the form was used to predict successful extubation (i.e., without adjustment; the scores were used to actually predict the results for the study group). In the ROC analysis, the AUC was found to be 0.898, and the optimal cut-off point was 5. The sensitivity, specificity, positive predictive value, negative predictive value, and accuracy were found to be 91.4%, 66.7%, 94.1%, 57.1%, and 87.8%, respectively. The results of the analyses are organized in Figure 4 and Table 5.

We used the six indicators in the extubation evaluation form as independent variables and successful extubation as the dependent variable. First, we used univariate logistic regression to conduct an analysis and discovered that only FiO_2_ ≤ 30% and PIP ≤ 15 cmH_2_O were significant; their crude OR values (95% CI) were 30.0 (2.87–313.47) and 8.0 (1.21–52.88), respectively. We then employed multivariate logistic regression and discovered that only FiO_2_ ≤ 30% remained significant; its adjusted OR (95% CI) was 30.0 (2.87–313.47), as presented in Table 6.

ROC analysis was conducted to determine the individual predictive power of the indicators on the extubation evaluation form. The results revealed that only FiO_2_ ≤ 30% and PIP ≤ 15 cmH_2_O exhibited an AUC greater than 0.7 (0.845 and 0.733, respectively) and exhibited high individual predictive power for successful extubation [17]. The other four indicators had AUCs smaller than 0.7; therefore, their individual predictive power was lower. The results are summarized in Table 7.

## 4. Discussion

A common method for evaluating the suitability of preterm infants for extubation is spontaneous breathing trials [2]. In this method, the mechanical ventilator setting is generally adjusted to only a 5–6 cmH_2_O continuous positive airway pressure. This is continued for 3–10 min, during which time clinicians observe for adverse clinical events, such as bradycardia or desaturation. Whether the trial is a success (i.e., whether the preterm infant is ready for extubation) is determined by the severity and duration of the adverse events. However, in this method, the mean airway pressure is reduced, and in preterm infants, for whom endotracheal tubes with a small inner diameter and large resistance are used, the spontaneous breathing trials may induce adverse clinical events [16].

In a systematic review and meta-analysis including patients with a birth weight below 1250 g and a gestational age of 24–34 weeks, spontaneous breathing trials were used to predict successful extubation. The results obtained indicated an overall sensitivity and specificity of spontaneous breathing trials of 95% (95% CI = 87–99%) and 62% (95% CI = 38–82%), respectively [5].

The present study used the extubation evaluation form to predict successful extubation for preterm infants with gestational age < 32 weeks and birth weight < 1500 g, with sensitivity and specificity of 91.4% and 66.7%, respectively. Compared with previous studies, the method adopted by the present study achieved similar predictive results for successful extubation without employing complex and risky spontaneous breathing trials. In addition, the readiness of intubated preterm infants for extubation can be assessed daily using the extubation evaluation form. Therefore, the form’s application will reduce the potential bias present in spontaneous breathing trials; generally, such trials are administered when a physician has determined that a patient may be ready for extubation. This may result in overestimated sensitivity and underestimated specificity [5].

During the ROC analysis, we determined that the AUC was 0.898, with an optimal cut-off point of 5. Thus, for preterm infant populations that meet inclusion criteria similar to those in this study, when such populations achieve a total score of 5 or higher on the extubation evaluation form, extubation may be considered.

The selection of the indicator “improved clinical condition” was based on the traditional approach of assessing the effectiveness of clinical treatment, which primarily focuses on the amelioration of preterm infants’ conditions necessitating medical intervention. This approach has gained widespread acceptance as the fundamental framework for evaluating the readiness for extubation in preterm infants [2,4].

The indicator named spontaneous breath rate ≥ 30 bpm was set based on the attending physician’s experience and the normal neonatal breath rate of 30–60 bpm [18]. We believe that the ability of preterm infants to breathe independently is a more important factor in determining readiness for extubation than simply referencing the respiratory rate setting on the mechanical ventilator. This is especially true since apneas are one of the most common causes of reintubation in this population [19,20].

The indicator named PIP ≤ 15 cmH_2_O was set by referencing the setting of mechanical ventilation before extubation in past research and physicians’ experiences. Previous studies have often regarded PIP ≤ 14–16 cmH_2_O as one of the requirements for the extubation of preterm infants [4,21,22]. Although more preterm infants met the PIP ≤ 15 cmH_2_O criterion in the ES group (than in the EF group) and had an AUC of 0.733 with regard to the prediction of successful extubation, this indicator was not statistically significant and was, thus, removed from the stepwise logistic regression model. This may be because the number of infants included in this study was small.

The indicator named FiO_2_ ≤ 30% was set by referencing the setting of mechanical ventilation before extubation in past research and physicians’ experiences [4]. In our study, we held the premise of maintaining SpO_2_ ≥ 90% while attempting to lower FiO_2_ [23]. The results of the multivariate logistic regression revealed that, among the six indicators used in this study, FiO_2_ ≤ 30% was the only indicator that remained significant, indicating that it is a crucial indicator in the evaluation of extubation suitability. Nevertheless, in clinical practice, no single indicator should be used alone to determine if the patient is ready for extubation. If FiO_2_ is used without consideration of the potential for spontaneous breathing or blood gas, the risk of extubation failure may substantially increase [24].

PvCO_2_ < 70 mmHg and blood pH value ≥ 7.2 were selected as indicators based on the strategy of permissive hypercapnia in the study. In the permissive hypercapnia ventilatory strategy, the ventilator is set at a reduced respiratory rate and tidal volume in the event of a lower pH (≥7.20) and higher PvCO_2_ (45–65 mmHg). This strategy can potentially reduce apnea and the duration of mechanical ventilation [25,26]. In this study, we increased the PvCO_2_ standard to <70 mmHg to facilitate earlier weaning of mechanical ventilation.

The four indicators in this study—namely improved clinical conditions, spontaneous breath rate ≥ 30 bpm, PvCO_2_ < 70 mmHg, and blood pH ≥ 7.2—when used alone did not demonstrate sufficient predictive power for successful extubation. This may be because the small sample size of this study provided insufficient evidence for predicting successful extubation solely through those four indicators. 

In the present study, compared with the ES group, the EF group had a lower birth weight, a longer duration of mechanical ventilation use, and a longer hospital stay. In addition, the EF group had a higher incidence of developing ROP grade 3 or higher and moderate or severe BPD. These phenomena were in line with those of past studies [9,27].

The study reported an extubation failure rate of 14.6% among preterm infants, indicating a substantial proportion experiencing extubation failure. Previous literature from 2014 documented a comparable population with an extubation failure rate of 25% ± 9% (mean ± standard deviation) [4]. More recent studies reported within the past two years, including those by Chen, Y.H., et al. (21.66%) [28], He, F., et al. (30.6%) [29], and Park, S.J., et al. (18.6%) [30], also reported extubation failure rates in similar populations. These findings underscore the continued significance of extubation failure rates in such preterm infant populations. Further investigation into the risk factors related to extubation failure and the ongoing development of enhanced assessment tools are necessary.

Our study had several limitations. One significant limitation was the small sample size. The number of infants included in this study was small, and the ES group was much larger than the EF group. The EF group comprised only six infants. This small number of intubated preterm infants is likely the result of respiratory treatment for preterm infants shifting toward the early use of noninvasive respiratory support [31,32]. Furthermore, the limited number of patients hindered the possibility of validation and precluded conducting subgroup analyses. To account for the small sample size, statistical analysis was performed using nonparametric methods in this study. We used the G*Power (version 3.1.9.4; Heinrich-Heine-Universität Düsseldorf, Düsseldorf, Germany) to perform calculations with the Wilcoxon–Mann–Whitney test and α error set at 0.05. The results show that despite the limited sample size, this research still achieves a statistical power of over 90%.

The assessment of the indicator “improved clinical condition” is performed by physicians and respiratory therapists with the aim of objectively evaluating the progression of preterm infants’ conditions. However, it is essential to acknowledge that, despite diligent efforts to uphold objectivity, a certain degree of subjectivity is inherent in this assessment.

Two of the indicators (PvCO_2_ < 70 mmHg and blood pH ≥ 7.2) were considered lenient, and few physicians would extubate preterm infants who had not achieved these standards. In addition, these two indicators are theoretically affected by the setting of PIP. When the measurements of these two indicators improve, PIP can be reduced. Therefore, regarding effectiveness, PIP may have a more profound influence.

Overall, the study results indicated that not all indicators on the extubation evaluation form were effective. Therefore, further adjustments are required and more cases should be included for analysis.

In our study, we did not assess the muscle strength of the preterm infants before extubation. It is thought that prolonged mechanical ventilation can lead to a decline in the strength of the respiratory muscles. However, research has shown that respiratory muscle strength is not a strong predictor of extubation success in preterm infants [33,34,35].

Studies have shown that it is reasonable to use steroids in infants at increased risk for airway edema and obstruction, such as those who have received repeated or prolonged intubations [36]. However, in this study, no steroids were used prior to extubation.

In recent years, an increasing number of studies have emphasized the value of lung ultrasound (LUS) in evaluating the severity and complications associated with RDS in preterm infants [37,38]. LUS exhibits remarkable sensitivity, specificity, and AUC when employed to predict the success of extubation in preterm infants with RDS [39,40]. However, the absence of relevant literature on the utilization of LUS for extubation prediction during the time of our study precluded its incorporation.

This study recruited preterm infants from the neonatal ICU of the Kaohsiung Medical University Chung-Ho Memorial Hospital with gestational age < 32 weeks and birth weight < 1500 g. Therefore, whether results of this study may be applied to older infants or those having undergone several intubations and whether the results may be applied to neonatal ICUs in other hospitals with different care methods and standards require further study.

## 5. Conclusions

In conclusion, the present study suggests that preterm infants who experience extubation failure may face prolonged mechanical ventilation and hospital stays, along with an increased risk of developing severe ROP and moderate-to-severe BPD. 

After adjusting for birth weight and postextubation respiratory device, the extubation evaluation form showed a significant association with successful extubation, indicating its potential as a predictive tool. However, further validation is needed to confirm the ability of the extubation evaluation form to predict successful extubation. These findings provide a foundation for future research on a larger scale.

## Figures and Tables

**Figure 1 children-10-01053-f001:**
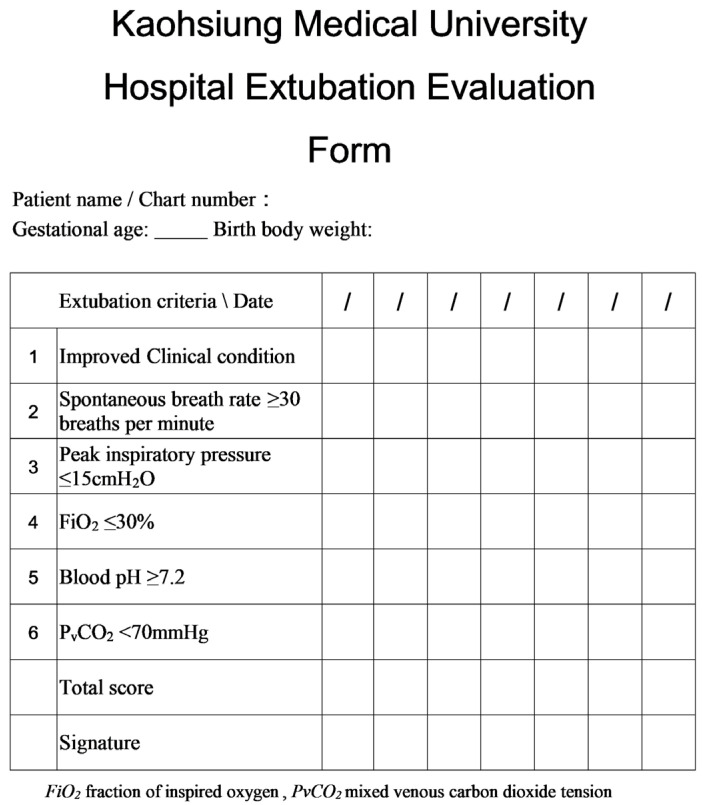
Extubation evaluation form.

**Figure 2 children-10-01053-f002:**
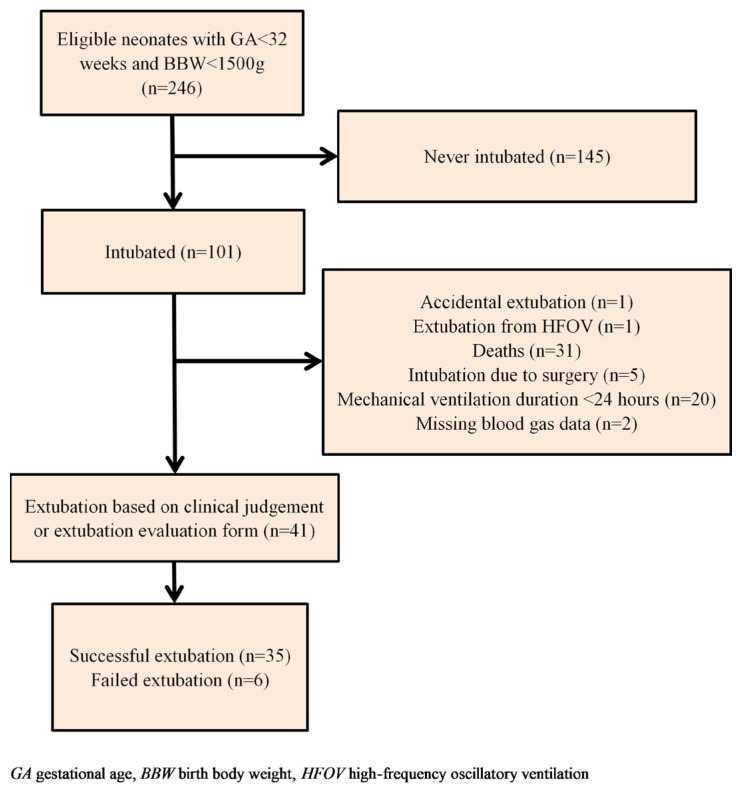
Flow diagram of preterm population in this study.

**Figure 3 children-10-01053-f003:**
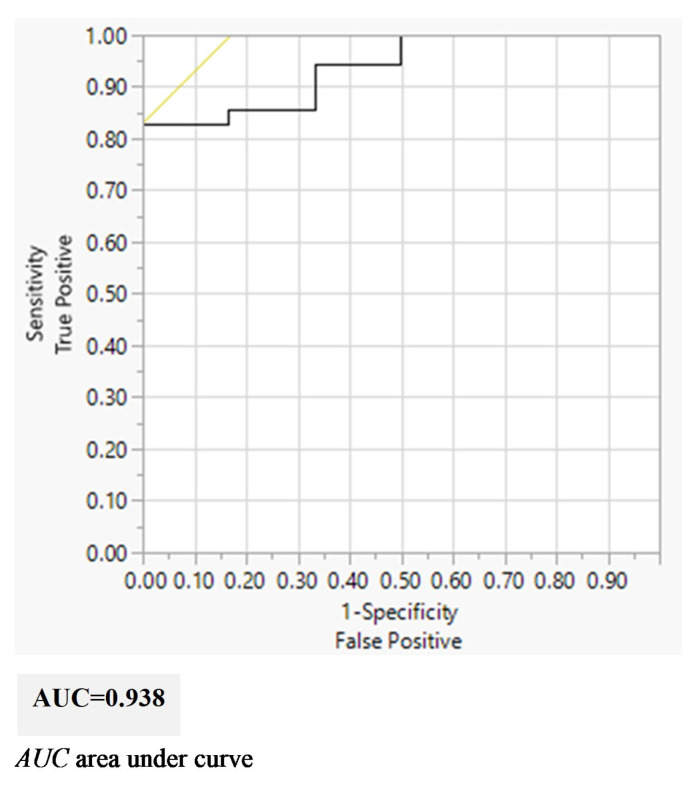
AUC of Logistic regression model for extubation evaluation form after adjustment.

**Figure 4 children-10-01053-f004:**
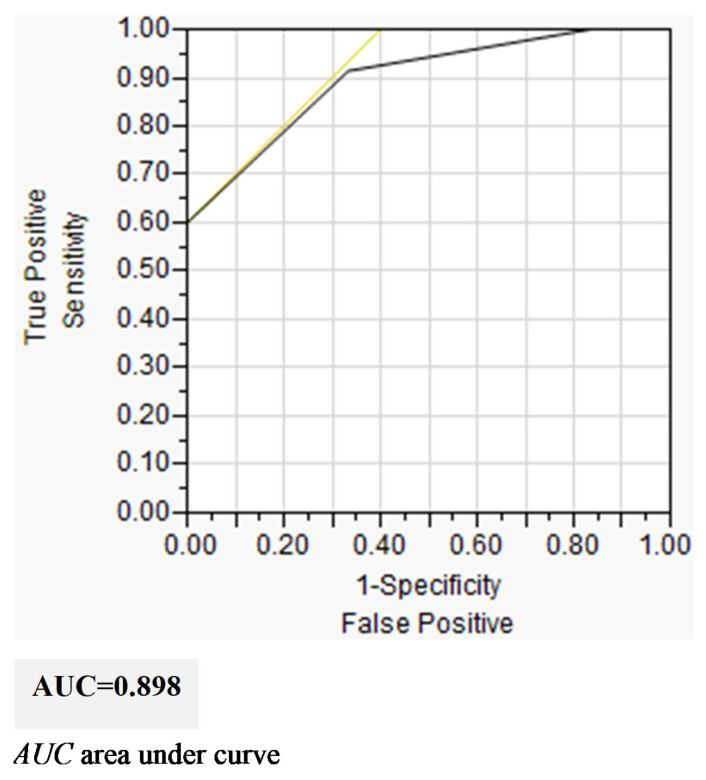
AUC of Extubation evaluation form to predict successful extubation.

**Table 1 children-10-01053-t001:** Characteristics between successful and failed extubation groups.

Clinical Variable	Extubation Success (n = 35)	Extubation Failure (n = 6)	*p* Value
Gestational age, wk	27 (25.9–28.4)	25.9 (25.3–26.9)	0.079
Birth weight, g	971 (848–1104)	750 (608–938)	0.015 *
Extremely preterm, No. (%)	24 (68.6)	6 (100.0)	0.167
Extremely low birth weight, No. (%)	20 (57.1)	6 (100.0)	0.07
Male, No. (%)	16 (45.7)	3 (50.0)	1
Antenatal steroids, No. (%)	26 (74.3)	3 (50.0)	0.334
Vaginal delivery, No. (%)	22 (62.9)	2 (33.3)	0.212
Apgar score at 5 min	6 (5–7)	5 (3–7)	0.369
Delivery room intubation, No. (%)	21 (60.0)	5 (83.3)	0.388
NTISS	28 (16–37)	21 (16–48)	0.825
Surfactant, No. (%)	30 (85.7)	5 (83.3)	1
Aminophylline or caffeine, No. (%)	33 (94.3)	6 (100.0)	1
Patent Ductus Arteriosus, No. (%)	28 (80.0)	5 (83.3)	1
Late onset sepsis, No. (%)	20 (57.1)	5 (83.3)	0.376
Postextubation NIPPV after first extubation attempt, No. (%)	14 (40.0)	4 (66.7)	0.377
Pre-extubation			
Days of life	6 (3–13)	16 (4–42)	0.363
Postmenstrual age, wk	28.3 (27–30.6)	28.6 (26.9–31.0)	0.81
Weight, g	952 (792–1187)	878 (734–1088)	0.417
Peak inspiratory pressure, cmH_2_O	13 (12–15)	16 (15–17)	0.006 *
FiO_2_, %	25 (21–30)	40 (37–51)	0.001 *
Set respiratory rate, bpm	25 (20–30)	40 (24–49)	0.042 *
pH	7.34 (7.33–7.41)	7.33 (7.23–7.40)	0.555
PvCO_2_, mmHg	44.1 (38.5–52.5)	47.0 (41.3–50.2)	0.519

Data are presented as median (interquartile range) unless otherwise indicated. NTISS, Neonatal Therapeutic Intervention Scoring System score; NIPPV, noninvasive positive pressure ventilation; FiO_2_, fraction of inspired oxygen; PvCO_2_, mixed venous carbon dioxide tension. * *p* value < 0.05.

**Table 2 children-10-01053-t002:** Outcomes between successful and failed extubation groups.

Clinical Variable	Extubation Success (n = 35)	Extubation Failure (n = 6)	*p* Value
Mechanical ventilation (via endotracheal tube) days before first extubation attempt	5 (2–10)	14 (3–28)	0.555
Cumulative mechanical ventilation (via endotracheal tube) days	5 (2–20)	51 (31–71)	0.002 *
Cumulative respiratory support device usage days ^1^	72 (55–92)	110 (88–162)	0.009 *
Cumulative ICU days	73 (59–100)	112 (96–155)	0.011 *
Cumulative hospital days	93 (71–112)	133 (108–162)	0.009 *
Complications			
Intraventricular hemorrhage, No. (%)	11 (31.4)	3 (50.0)	0.393
Retinopathy of prematurity, No. (%)	20 (57.1)	6 (100.0)	0.07
Retinopathy of prematurity ≥ grade 3, No. (%)	3 (8.6)	5 (83.3)	<0.001 *
Retinopathy of prematurity that needed laser therapy, No. (%)	5 (14.3)	4 (66.7)	0.015 *
Bronchopulmonary dysplasia, No. (%)	33 (94.3)	6 (100.0)	1
Moderate or severeBronchopulmonary dysplasia, No. (%)	14 (40.0)	6 (100.0)	0.009 *

Data are presented as median (interquartile range) unless otherwise indicated. * *p* value < 0.05. ^1^ Respiratory support device includes invasive positive pressure ventilation, noninvasive positive pressure ventilation, nasal continuous positive airway pressure, and high-flow nasal cannula.

**Table 3 children-10-01053-t003:** Extubation evaluation form data between successful and failed extubation groups.

Extubation Evaluation Form Indicator	Extubation Success (n = 35)	Extubation Failure (n = 6)	*p* Value
FiO_2_ ≤ 30%, No. (%)	30 (85.7)	1 (16.7)	0.002 *
Peak inspiratory pressure ≤ 15 cmH_2_O, No. (%)	28 (80.0)	2 (33.3)	0.035 *
Improvement of the clinical condition, No. (%)	30 (85.7)	3 (50.0)	0.077
Blood pH ≥ 7.2, No. (%)	35 (100.0)	5 (83.3)	0.146
Spontaneous breath rate ≥ 30 breaths per minute, No. (%)	34 (97.1)	6 (100.0)	1
P_v_CO_2_ < 70 mmHg, No. (%)	35 (100.0)	6 (100.0)	-
Total score, Median (IQR)	6 (5–6)	4 (3–5)	0.001 *

FiO_2_, fraction of inspired oxygen; PvCO_2_, mixed venous carbon dioxide tension. * *p* value < 0.05.

**Table 4 children-10-01053-t004:** Logistic regression model for extubation evaluation form after adjustment.

Term	Odds Ratio	95% Confidence Intervals (Lower, Upper)		*p* Value
Extubation evaluation form score	6.649	1.297	34.075	0.023
Birth weight in g	1.006	0.999	1.013	0.072
Postextubation device (NIPPV)	0.644	0.048	8.622	0.739

NIPPV, noninvasive positive pressure ventilation.

**Table 5 children-10-01053-t005:** ROC table of extubation evaluation form to predict successful extubation.

Score	Sensitivity	Specificity	PPV	NPV	Accuracy
6	60.0%	100%	100%	30.0%	65.9%
5 ^1^	91.4%	66.7%	94.1%	57.1%	87.8%
4	97.1%	33.4%	89.5%	66.7%	87.8%
3	100%	16.7%	87.5%	100%	87.8%
2	100%	0%	85.4%	-	85.4%
1	-	-	-	-	85.4%

ROC, receiver operating characteristic; PPV, positive predictive value; NPV, negative predictive value. ^1^ Cut-off point.

**Table 6 children-10-01053-t006:** Logistic regression of each extubation evaluation form indicator for successful extubation.

Extubation Evaluation Form Indicator	Extubation Success (n = 35)	Extubation Failure (n = 6)	Crude OR (95% CI)	*p* Value	Adjusted OR (95% CI)	*p* Value
FiO_2_ ≤ 30%	30 (85.7)	1 (16.7)	30.00 (2.87–313.47)	0.004	30.00 (2.87–313.47)	0.004
Peak inspiratory pressure ≤ 15 cmH_2_O	28 (80.0)	2 (33.3)	8.00 (1.21–52.88)	0.031	-	-
Improvement of the clinical condition	30 (85.7)	3 (50.0)	6.00 (0.94–38.52)	0.059	-	-
Blood pH ≥ 7.2	35 (100.0)	5 (83.3)	N.A.	1	-	-
Spontaneous breath rate ≥ 30 breaths per minute	34 (97.1)	6 (100.0)	N.A.	1	-	-
PvCO_2_ < 70 mmHg	35 (100.0)	6 (100.0)	N.A.	-	-	-

The adjusted OR and 95% CI were estimated by a stepwise logistic regression method; the significant indicators were entered into this model (*p* < 0.05), otherwise indicated by ‘–’. FiO_2_, fraction of inspired oxygen; PvCO_2_, mixed venous carbon dioxide tension; OR, odds ratio; CI, confidence interval.

**Table 7 children-10-01053-t007:** ROC of each extubation evaluation form indicator for successful extubation.

Extubation Evaluation Form Indicator	AUC
FiO_2_ ≤ 30%	0.845
Peak inspiratory pressure ≤ 15 cmH_2_O	0.733
Improvement of the clinical condition	0.679
Blood pH ≥ 7.2	0.583
PvCO_2_ < 70 mmHg	0.500
Spontaneous breath rate ≥ 30 breaths per minute	0.486

ROC, receiver operating characteristic; AUC, area under curve; FiO_2_, fraction of inspired oxygen; PvCO_2_, mixed venous carbon dioxide tension.

## Data Availability

The data used to support the findings of this study are available from the corresponding author upon request.

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
