# Peer review of "Feasibility of Endotracheal Extubation Evaluation Form in Predicting Successful Extubation in Neonatal Intensive Care Units: A Retrospective Study"

_children, 2023, doi:10.3390/children10061053_

Round 1
Reviewer 1 Report
This paper aims to establish a scoring method that contributes to the success of extubation.They concluded that the total score of the form was significantly associated with successful extubation; the higher the score, the greater the chance of successful extubation, the extubation evaluation form may provide a more objective standard for extubation assessment in preterm infants. There are many deficiencies in this paper:1)The sample size of the included studies was small. 2)According to the scoring criteria provided by this study, 17% of the children still failed to wean, suggesting that the success rate of weaning was not high.
Author Response
Response to Reviewer 1 Comments
Point 1: The sample size of the included studies was small.
Response 1: As our study adopts a retrospective design, determining an optimal sample size posed a challenge. An important limitation of our study was the relatively small number of infants included, despite the extensive retrospective period of 5 years and 4 months. This limited cohort of intubated preterm infants is plausibly attributed to the progressive shift in respiratory treatment practices towards early implementation of noninvasive respiratory support.
To account for the small sample size, statistical analysis was performed using nonparametric methods in this study. We used the G*Power software to perform calculations with the Wilcoxon-Mann-Whitney test and α error set at 0.05. The results show that despite the limited sample size, this research still achieves a statistical power of over 90%.
Point 2: According to the scoring criteria provided by this study, 17% of the children still failed to wean, suggesting that the success rate of weaning was not high.
Response 2: Thank you for your advice, but there might be some misunderstandings in it. The study enrolled a total of 41 preterm infants, of which 35 had successful extubation and 6 had failed extubation. Therefore, the extubation failure rate should be 6/41=14.6%. According to a systematic review by Annie Giaccone in 2014, the re-intubation rate in similar population of preterm infants was 25% ± 9% (mean ± SD) [1]. Therefore, we believe that the extubation success rate in this study may not be considered low.
- 1. Giaccone, A.; Jensen, E.; Davis, P.; Schmidt, B. Definitions of extubation success in very premature infants: A systematic re-view. Arch. Dis. Child. Fetal Neonatal Ed. 2014, 99, F124–F127. doi: 10.1136/archdischild-2013-304896.

Reviewer 2 Report
Dear Author,
In this manuscript, you analyzed the relevant factors in the successful extubation of preterm neonates and determined whether the developed extubation assessment form can effectively predict extubation.
Introduction:
The introduction is well organized and presents relevant information for the content of the article.
Materials and Methods: are well structured, with an adequate description of study design. . However, a small number of patients enrolled in the study considering the duration of the study over 5 years and 4 months (January 1, 2014, to April 24, 2019).
Results and discussion: As you mentioned, the study had limitations. The number of infants included in this study was small with the big difference regarding the number of newborns between the two groups (ES vs EF group). This study is a beginning in finding a formula for evaluating extubation. It is necessary to conduct a study on a larger number of newborns and possibly include some new parameters.
On the other hand the discussions are well structured, using also references to the bibliography.
The conclusions of this study are presented in accordance with the topic analyzed.
It is a well written article.
Author Response
Response to Reviewer 2 Comments
Point 1: However, a small number of patients enrolled in the study considering the duration of the study over 5 years and 4 months (January 1, 2014, to April 24, 2019).
Response 1: As our study adopts a retrospective design, determining an optimal sample size posed a challenge. An important limitation of our study was the relatively small number of infants included, despite the extensive retrospective period of 5 years and 4 months. This limited cohort of intubated preterm infants is plausibly attributed to the progressive shift in respiratory treatment practices towards early implementation of noninvasive respiratory support.
To account for the small sample size, statistical analysis was performed using nonparametric methods in this study. We used the G*Power software to perform calculations with the Wilcoxon-Mann-Whitney test and α error set at 0.05. The results show that despite the limited sample size, this research still achieves a statistical power of over 90%.

Reviewer 3 Report
The paper is interesting from a practical point of view. I have no comments on the methodology of the study. However, I would suggest that the conclusions be changed. Conclusions should be general in nature represent a generalization and not a re-iteration of the results. Please consider expanding the paragraph limitations to include a wide range of factors that may have influenced the results obtained.
Author Response
Response to Reviewer 3 Comments
Point 1: However, I would suggest that the conclusions be changed. Conclusions should be general in nature represent a generalization and not a re-iteration of the results.
Response 1: Thank you for your advice; we have revised the content based on your advice.
Point 2: Please consider expanding the paragraph limitations to include a wide range of factors that may have influenced the results obtained.
Response 2: Thank you for your advice; we have revised the content based on your advice.

Round 2
Reviewer 1 Report
Although there is evidence of a higher rate of extubation failure in the past, the literature is from 10 years ago and cannot be compared with 10 years later. In addition, note that guiding extubation under lung ultrasound monitoring can almost avoid extubation failure【In addition, note that guiding extubation under lung ultrasound monitoring can almost avoid extubation failure.In addition, note that guiding extubation under lung ultrasound monitoring can almost avoid extubation failure】. Therefore, this paper has major limitations, including the small sample size. All of this needs to be made clear in terms of limitations.
Author Response
Response to Reviewer 1 Comments
Point 1: Although there is evidence of a higher rate of extubation failure in the past, the literature is from 10 years ago and cannot be compared with 10 years later.
Response 1: Thank you for your valuable feedback, and we appreciate your observation that the previous systematic review provided was outdated. Given this concern, we made diligent efforts within the limited time available to identify studies that encompassed population characteristics and extubation criteria resembling those in our research. Regrettably, our search did not yield any recent systematic reviews addressing this specific topic. Instead, we found a limited number of articles that partially matched the conditions for comparison.
In 2022, Chen, Y.H., et al. reported a study involving 60 infants with birth weight less than 1500 gm who underwent their initial extubation in NICU. Successful extubation was defined as the absence of reintubation within a 3-day period following extubation. Out of the 60 infants, 13 (21.66%) failed extubation [1].
In 2022, He, F., et al. reported a retrospective study involving extremely and very preterm infants born at gestational ages of less than 32 weeks. Extubation failure was defined as the requirement for reintubation within 72 hours following extubation. A total of 359 infants were included in the study. Among these infants, there were 249 successful extubation cases (69.4%) and 110 failed cases (30.6%). The observed failure rate of 30% in this study was higher compared to findings reported in other studies, suggesting a potential disparity in the treatment approach at the center where the study was conducted [2].
In 2023, Park, S.J., et al. reported a study focusing on preterm infants born at gestational ages of less than 32 weeks. Extubation failure (EF) was defined as the requirement for intubation within 7 days following extubation. 129 infants were included. The EF rate in this study was determined to be 18.6% (24/129). Notably, approximately 80% of patients in the EF group required re-intubation within an average time of 90.17 hours [3].
(It can be inferred that approximately 14.9% of preterm infants require reintubation within an average time of 90.17 hours.)
Based on the aforementioned articles, we can infer that there are significant variations in extubation failure rates among different research institutions and care approaches. Due to the limited availability of recent large-scale studies, it remains challenging to determine an acceptable extubation failure rate that is comparable to our study population and defined criteria. However, it is undeniable that in such a population of premature infants, the rate of extubation failure is indeed high.
We have incorporated your suggestions and made revisions accordingly. (Page 11, line 320.)
- Chen, Y.H., et al., Analysis of predictive parameters for extubation in very low birth weight preterm infants. Pediatr Neonatol, 2023. 64(3): p. 274-279.
- He, F., et al., Predictors of extubation outcomes among extremely and very preterm infants: a retrospective cohort study. J Pediatr (Rio J), 2022. 98(6): p. 648-654.
- Park, S.J., et al., Risk factors and clinical outcomes of extubation failure in very early preterm infants: a single-center cohort study. BMC Pediatr, 2023. 23(1): p. 36.
Point 2: In addition, note that guiding extubation under lung ultrasound monitoring can almost avoid extubation failure.
Response 2: Thank you for your feedback; we have incorporated your suggestions and made revisions accordingly. (Page 12, line 361.)
Point 3: Therefore, this paper has major limitations, including the small sample size. All of this needs to be made clear in terms of limitations.
Response 3: Thank you for your feedback; we have incorporated your suggestions and made revisions accordingly. (Page 11, line 330.)
